# A Meta-Analysis on Quantitative Calcium, Phosphorus and Magnesium Metabolism in Horses and Ponies

**DOI:** 10.3390/ani14192765

**Published:** 2024-09-25

**Authors:** Isabelle Maier, Ellen Kienzle

**Affiliations:** Department of Veterinary Sciences, Ludwig-Maximilians-Universität München, Schoenleutnerstr. 8, D-85764 Oberschleissheim, Germany; kienzle@tiph.vetmed.uni-muenchen.de

**Keywords:** pony, horse, mineral, metabolism, calcium, phosphorus, magnesium, digestibility

## Abstract

**Simple Summary:**

The present study evaluated the literature investigating the potential differences in the quantitative calcium (Ca), phosphorus (P) and magnesium (Mg) metabolism in horses and ponies and between “organic” (plant origin) and “inorganic” mineral sources (mineral salts). For P sources, the “inorganic” P salts were also differentiated according to water solubility. The present study found unequivocal differences in apparent Mg digestibility between horses and ponies, whereby horses require a greater amount of this nutrient. “Organic” Ca was shown to have a higher bioavailability than “inorganic” Ca. When considering P sources, the distinction was made between water-soluble “inorganic” sources and all other sources. The water-soluble sources were highly available, and they increased serum P levels and renal P excretion, which presents a potential health risk.

**Abstract:**

The aims of the present meta-analysis were (i) to re-evaluate the factorially calculated Ca, P and Mg requirements to replace endogenous faecal losses, taking new data into account, (ii) to identify potential differences between horses and ponies regarding requirements, apparent digestibility, serum levels and renal excretion of Ca, P and Mg and (iii) to investigate the influence of mineral sources, i.e., “inorganic” sources from added mineral salts and “organic” sources from feed plants. For P, the water solubility of “inorganic” sources was taken into consideration. Data on the aforementioned parameters from 42 studies were plotted against intake, similar to the Lucas test for true digestibility and faecal endogenous losses. Within specific intake ranges, data were compared using *t*-tests and an ANOVA, followed by Holm–Sidak post hoc tests. Ponies had lower endogenous faecal Mg losses than horses. Consequently, apparent Mg digestibility was higher in ponies. Factorial calculations of Mg requirements to replace faecal losses showed that ponies needed approximately half of the current recommended amount, while horses required 1.9 times the amount currently recommended by Kienzle and Burger. The overall mean matched previous recommendations. For Ca, there was no discernible difference between ponies and horses. True Ca digestibility calculated by the Lucas test was higher and endogenous losses were lower when “organic” Ca was fed as opposed to when “inorganic” sources were used. The resulting factorial calculations of the requirements to replace faecal losses were close to current recommendations for “organic” Ca. For “inorganic” sources, however, the new calculations were below the recommended level. For P, there were no discernible differences between horses and ponies. There were also no clear effects of “inorganic” or “organic” P sources. The water solubility of “inorganic” sources was the key factor determining P metabolism. Water-soluble P sources exhibited higher true and apparent digestibility. The intake of these P sources led to hyperphosphatemia and hyperphosphaturia, even at low intakes. In other species, this has been shown to pose a health risk. Therefore, it is recommended to avoid the use of highly water-soluble “inorganic” P sources in horses and ponies. Given the lower digestibility of insoluble P sources, the factorially calculated P requirements for such sources are higher than the current recommendations.

## 1. Introduction

Ca, P and Mg are essential elements that are crucial for many bodily functions. Apparent and true digestibility as well as endogenous faecal losses of these minerals are important for factorial calculation of mineral requirements. For adult horses, a meta-analysis on these parameters was last performed in 2011 [1]. Since this time, the data available in the literature have increased. Therefore, it is possible to recalculate the results of Kienzle and Burger [1] with a larger dataset. This was the first goal of the present meta-analysis.

A more comprehensive database provides the possibility to investigate potential differences between horses and ponies. This was the second goal of the present meta-analysis.

The third question addressed was the difference between mineral sources, especially P sources. In other species such as cats, dogs, pigs and humans, it has been shown that the P source may play an important role in digestibility and metabolism [2,3,4,5]. “Inorganic” P is even considered to be a health risk [6,7,8,9,10,11,12,13,14,15]. In this context, renal P excretion and serum P levels play a crucial role [6,16,17,18,19,20]. Therefore, data on serum minerals, renal mineral excretion and retention were included in the meta-analysis, covering not only P but also Ca and Mg.

Plots similar to the Lucas test were used as the main approach. When apparent digestibility, faecal excretion or apparently digested minerals are plotted against mineral intake, patterns may emerge that are not visible when only comparing apparent digestibility [21,22,23]. The same is true for serum levels and renal excretion if plotted against intake.

## 2. Materials and Methods

The studies used for the present meta-analysis were sought using Google Scholar, the database information system (DBIS) of the LMU Munich, PubMed and the Online Public Access Catalogue (OPAC) of the LMU Munich and the Bavarian State Library (BSB). The main keywords searched for were “mineral digestibility, calcium, phosphorus, magnesium, horse, pony, mare, gelding, digestibility, availability, minerals, set elements, renal, kidney, faecal, serum, blood, excretion, resorption or absorption” in various configurations. Theses that were used in the meta-analysis of Kienzle and Burger [1] were also included. Only data from studies that involved adult equines (older than 36 months following the Society of Nutrition Physiology (GfE) [24]), which were neither pregnant nor lactating, were used. Since the loss of the minerals Ca, P and Mg through sweat is minimal in horses and ponies [25] and does not lead to a recommendation for additional intake beyond the maintenance requirements according to GfE [24], both working and non-working animals were included in the analysis.

Only studies from which at least two of the following parameters could be extracted were considered: daily mineral intake, apparent digestibility, faecal excretion, renal excretion and serum blood levels. The body weight of the animals used in the studies had to be available, or at least the breed had to be stated, to estimate the body weight. This calculation was made based on the body mass in kg of large adult horses with a body condition score (BCS) of 5–6 [24]. In addition, the parameters of breed, age, feed composition, mineral source, dry matter intake, dry matter digestibility and faecal dry matter excretion were required. Both individual data and group averages were taken into account.

The present meta-analysis was conducted in accordance with the Systematic Reviews and Meta-Analyses (PRISMA) statement [26]. The last search for sources was carried out in July 2024. The following studies were used: [27,28,29,30,31,32,33,34,35,36,37,38,39,40,41,42,43,44,45,46,47,48,49,50,51,52,53,54,55,56,57,58,59,60,61,62,63,64,65,66,67,68]. We included in the Appendix A a flowchart in Appendix A for the study search and selection and also Appendix A, which identifies the number of studies and participating individuals for each graph.

Classification as a pony or horse was determined by body weight: animals with a body weight of more than 300 kg were considered horses, and those with a body weight of less than 300 kg were considered ponies. In a study conducted by the Chair of Animal Nutrition and Dietetics in Munich, one pony was temporarily heavier than 300 kg (ad libitum feeding). Nevertheless, the pony was still counted as a pony in the respective study.

Ca, P and Mg were subdivided depending on “organic” or “inorganic” origin. The term “organic” does not refer to the chemical definition but rather describes the natural mineral content of plants, such as Ca from alfalfa. This does not exclude the possibility that the mineral in the plants could be an “inorganic” compound in the chemical sense. In contrast, “inorganic” refers to mineral compounds added to mineral supplements or mixed feed. The terms “organic” and “inorganic” have been widely used in this context in various species [4,5,19,20]. To avoid confusion with chemical definitions, the term is placed in quotation marks. One study by Schulze [58] could not distinguish between sources, because the information about the feed did not allow for the discrimination of “organic” vs. “inorganic” origins, and was not used in this context.

The recommendations on requirements of GfE [24] were used to compare intake to requirements. The reference range for serum mineral level was taken from the book *Equine Applied and Clinical Nutrition: Health, Welfare and Performance* [25].

For the meta-analysis, all data on intake and excretion were calculated per kilogram of metabolic body weight (MBW). The initial step in the evaluation involved plotting apparent digestibility against intake. The goal was to assess whether a typical hyperbolic curve appeared, as proof of the validity of data from different sources. To compare data and reveal potential interactions, the so-called Lucas test or modified Lucas test is an effective method [1,69]. That was the second step. In the Lucas test, the apparently digested amount of a nutrient is plotted against its intake. Provided that the data distribution is suitable for linear regression calculation, the regression coefficient multiplied by 100 will represent the true digestibility (in %) and the intercept will represent the endogenous losses. This version of the Lucas test is most applicable to nutrients with high apparent digestibility, such as Ca. In the modified Lucas test, faecal nutrient excretion is plotted against intake. In this case, the regression coefficient must be subtracted from 1 and then multiplied by 100 to calculate true digestibility. This version is suitable for nutrients with a lower apparent digestibility such as P and Mg in the present study. The mineral requirement to replace faecal losses is then calculated by dividing the endogenous losses by the true digestibility and multiplying by 100. Data on serum mineral concentration were plotted against mineral intake. Data on renal excretion were plotted against intake and/or apparently digested amount of the respective mineral. Mineral retention was plotted against mineral intake. Outliers were present in almost all plots. All outliers were examined for causes of exceptional mineral metabolism, such as extremely high aluminium intake in the study by Roose [51], which affected serum Ca and renal P excretion. The data of Lensing [70] created outliers in practically all diagrams, a finding that clearly suggests that the data are not suitable for the present meta-analysis. In that study, nutrient supply was changed in rapid succession with short wash-out periods. Therefore, data from the aforementioned study were not used.

Acid–base balance has well-known effects on Ca and P metabolism [28,30,59,67,71,72]. Therefore, data from studies involving shifts in acid–base balance were excluded from all plots on renal Ca and P excretion and retention. For serum Ca, data from studies with shifts in acid–base balance fell outside the range defined by three standard deviations and were therefore not included in the analyses of serum Ca. In detail, the aforementioned limitations pertained to the following studies: Baker [27,28], O’Connor [47], McKenzie [42], Berchtold [30], Mueller [43], Stürmer [59], Wall [67], Schryver [57], one trial of Gomda [34] and one trial of Mundt [44]. In the plots of apparent digestibility, faecal excretion and apparently digested mineral, these datasets were included because they did not exhibit any anomalies.

The Ca:P ratio in food has a strong and well-known effect on renal Ca and P excretion [31,61,66,73]. Data from studies with very low or extremely high Ca:P ratios were therefore not included in the present meta-analysis. Only studies with a Ca:P ratio between 1:1 to 5:1 were included in the analysis of renal Ca and P excretion. Again, in the plots of apparent digestibility, faecal excretion and apparently digested mineral, these datasets were included because they did not exhibit any anomalies.

For Ca intakes exceeding 1000 mg/kg MBW, data were only available from ponies fed “inorganic” Ca sources. Consequently, the diagrams comparing “organic” and “inorganic” Ca sources—regarding the apparently digested Ca quantity, renal Ca excretion and Ca retention—were limited to an intake of up to 1000 mg/kg MBW.

The plots of renal P excretion and retention in relation to P intake were restricted to an intake of up to 500 mg/kg MBW because data above this level were limited to ponies consuming “inorganic” P.

The data of Baker [27] were eliminated from the Mg serum level plot, as they produced outliers. The Mg serum level in this study was in a range that could have caused clinical problems. Presumably, there is an error in units somewhere.

The comparison of two means was conducted by Student’s *t*-test or alternatively, if data were not normally distributed, by a Mann–Whitney Rank Sum Test. If multiple influencing factors were present, a two-way ANOVA was used, followed by the Holm–Sidak post hoc test for all pairwise comparisons. These tests were performed with SigmaPlot 14 (Systat Software, San Jose, CA, USA). Hyperbolic regressions were calculated in SigmaPlot as nonlinear inverse first-order regressions. Linear regressions were also determined in SigmaPlot. Linear regression lines were compared using the BiAS program with the test of Ho (BiAS. für Windows, Version 11.01, 2023, epsilon-Verlag, Frankfurt, Germany). The significance level was defined at <0.05.

## 3. Results

### 3.1. Calcium

#### 3.1.1. Ca Digestibility

Figure 1 shows the apparent digestibility of Ca in relation to Ca intake. Especially when fed below the required levels, apparent digestibility was low, and in some cases, even negative. This effect appeared to be stronger in horses than in ponies. Within this intake range, the Ca sources were typically “organic” from feed. The GfE recommendation for Ca supply for maintenance is 164 mg/kg MBW. The mean apparent digestibility of Ca at an intake of less than 164 mg/kg MBW of Ca was significantly higher in ponies than in horses. At higher intake levels, there was no difference between ponies and horses.

Figure 2 shows the apparently digested Ca in relation to Ca intake. Trendlines mark the “organic” and “inorganic” sources. The regression coefficient was higher for “organic” sources. A two-factorial ANOVA of apparent Ca digestibility within the intake range of 164–1000 mg/kg MBW, considering the factors pony vs. horse and “organic” vs. “inorganic”, showed a significant effect of the Ca source. Specifically, “organic” Ca had higher values compared to “inorganic” Ca (“organic”: 48 ± 10.3%, n: 127; “inorganic”: 39 ± 18.4%, n: 163; *p* < 0.001). No significant effect of pony vs. horse was observed. There was no significant interaction between the factors (*p* = 0.143).

#### 3.1.2. Serum Ca Concentration

Within the intake range of 164 to 1000 mg/kg MBW, the serum Ca concentration (Figure 3) was completely unaffected by Ca intake, source, Ca:P ratio or whether the subject was a pony or a horse.

#### 3.1.3. Ca Renal Excretion

Renal excretion was investigated only in the range of a Ca:P ratio between 1:1 and 5:1 (Figure 4). Experiments with shifts in acid–base balance were not considered. A two-way ANOVA of renal Ca excretion as a percentage of apparently digested Ca at an intake of 164 to 1000 mg/kg MBW showed no significant difference between the factors pony and horse nor between “organic” and “inorganic” Ca sources. There was no significant interaction between pony vs. horse and “organic” vs. “inorganic”.

#### 3.1.4. Ca Retention

Retention of Ca (Figure 5) increased with higher intake levels. In ponies, the retention seemed to be higher than in horses. A two-way ANOVA of Ca retention as a percentage of intake with the factors horse vs. pony and “organic” vs. “inorganic” Ca source showed a significant difference between ponies and horses (pony: mean = 22.8%; horse: mean = 8.5%). There was no effect between the different Ca sources (“organic” Ca: mean = 15.7%; “inorganic” Ca: mean = 15.5%). The retention in percent of intake averaged 16%.

### 3.2. Phosphorus

#### 3.2.1. P Digestibility

Figure 6 illustrates the apparent digestibility of P in relation to intake. There was neither a pronounced difference between pony and horse nor between “organic” and “inorganic” P sources. When “inorganic” P sources were split into water-soluble (Na_2_HPO_4_, NaH_2_PO_4_ or Ca(H_2_PO_4_)_2_) and other “inorganic” sources (such as CaHPO_4_), a significant difference was revealed. A statistical comparison between the apparent digestibility of water-soluble “inorganic” P sources and all other P sources in the range above the GfE [24] recommended requirements of 113 mg/kg MBW revealed a significantly higher apparent digestibility of water-soluble “inorganic” P sources (median soluble “inorganic” P: 20.3%; median all other P sources: 8.2%; *p* < 0.001).

When faecal P excretion was plotted against P intake (Figure 7), there was no apparent effect of either being a pony or a horse nor of the P source (“organic” vs. “inorganic”). Trendlines were calculated separately for ponies and horses, as well as for “organic” vs. “inorganic” sources. There was no significant difference between the pony and horse trendlines (test from Ho, *p* = 0.541). The data were again divided based on the water solubility of “inorganic” sources compared to all other sources (Figure 8). Faecal P excretion in relation to intake was lowest when water-soluble “inorganic” sources were used. The faecal excretion was clearly lower than for all other P sources, especially if intake was high. The relationship between intake and faecal excretion was linear for all sources except for the soluble “inorganic” sources. Here, the regression equation was curvilinear.

#### 3.2.2. Serum P Concentration

Figure 9 shows the P serum levels. P serum levels above the reference range were observed only for water-soluble “inorganic” sources.

#### 3.2.3. P Renal Excretion

Renal P excretion in relation to P intake up to an intake of 500 mg/kg MBW is shown in Figure 10. Experiments with acidifying rations were excluded. Again, the data were split according to solubility into highly water-soluble “inorganic” sources (Na_2_HPO_4_, NaH_2_PO_4_ or Ca(H_2_PO_4_)_2_) and other P sources (Figure 11). The water-soluble “inorganic” P sources led to a high renal P excretion. A median of about 32% of the apparently digested P quantity was excreted renally when highly water-soluble sources were used. In comparison, the use of other P sources resulted in a renal P excretion of approximately 4% of the apparently digested P amount. The difference was highly significant.

#### 3.2.4. P Retention

Figure 12 shows the P retention in relation to P intake. Data distribution did not allow for a conclusive evaluation with regard to ponies and horses. There was, however, a difference between “inorganic” and “organic” P sources (Figure 12). “Inorganic” P tended to result in lower P retention than “organic” P sources, especially at high intake. The average retention, expressed as a percentage of intake, was 5.9%.

### 3.3. Magnesium

#### 3.3.1. Mg Digestibility

The apparent digestibility of Mg in relation to intake is shown in Figure 13. In horses, negative apparent digestibility was observed even at an intake of up to 120 mg/kg MBW. In ponies, the apparent digestibility was mainly positive across the entire range of intake. In the range of 53 mg/kg MBW (recommended requirement for maintenance following GfE [24]) to 200 mg/kg MBW, the apparent digestibility was significantly higher in ponies than in horses (*p* = 0.001). There was no effect of the Mg source.

Figure 14 shows the faecal Mg excretion in relation to Mg intake. Trendlines mark the values for ponies and for horses. The intercept, i.e., the endogenous faecal excretion, was significantly higher in horses than in ponies (horse: 95% CI [6.66, 26.27); pony: 95% CI [2.04, 7.91]; *p* < 0.001). The regression coefficients were similar.

#### 3.3.2. Serum Mg Concentration

In Figure 15, the serum Mg level is plotted against Mg intake. With the exception of the study of Pferdekamp [50], there was a significant increase in serum Mg concentration with increasing intake. Pferdekamp [50], however, used a ration with exceptionally high Ca and P intake. For comparison, the median Ca intake in the aforementioned study was 1771.36 mg/kg MBW, whereas the median Ca intake in the other studies referenced in the figure was 361.27 mg/kg MBW. Due to the data distribution, differences between horses and ponies or “organic” and “inorganic” sources could not be further investigated.

#### 3.3.3. Mg Renal Excretion

Figure 16 shows the renal Mg excretion in relation to the amount of apparently digested Mg. The Mg intake was not limited. The data distribution with regard to source, pony or horse, and the amount of apparently digested Mg did not permit a conclusive evaluation of these factors. There was a general trend towards higher renal Mg excretion with increasing amounts of apparently digested Mg.

#### 3.3.4. Mg Retention

The retention of Mg in relation to Mg intake is shown in Figure 17. The data distribution did not allow for statistical evaluation. The retention as a percentage of intake showed no obvious differences between ponies and horses, or between “inorganic” and “organic” sources. The mean retention was 0.7%.

## 4. Discussion

### 4.1. The Lucas Test

Apparent digestibility is a relative measurement identifying the percentage of the intake of a nutrient which is not excreted via the faeces. Endogenous faecal losses are not considered. Although this mathematical description is often very useful, it may also obscure differences in results between experiments conducted under varying conditions, particularly differing intake levels. For instance, if the intake is very low, apparent digestibility may be negative even though a considerable part of the nutrient from the feed is absorbed. This is due to endogenous losses exceeding the intake, which is shown very clearly in Figure 1, Figure 6 and Figure 13, depicting apparent digestibility plotted against intake. The Lucas test or modified Lucas test is a very valuable tool to combine data from the literature and make quantitative differences or similarities in digestion more visible [1,21,74,75]. The amount of an apparently digested nutrient is plotted against intake for nutrients with relatively high apparent digestibility. For nutrients with a low apparent digestibility, intake is plotted against faecal excretion. Similar effects can be observed when renal excretion or serum concentration is plotted against intake or the apparently digested amount.

In the present study, these techniques were used. Outliers were present in most plots, and the data were reviewed to determine the reasons for these deviations. The most conspicuous study was by Lensing [70]. The data in this study produced outliers in almost all parameters related to mineral excretion, regardless of the mineral. In this study, the animals were fed a surplus or deficit of Ca and an excess of P and vitamins A and D in short experimental periods (50, 32 and 30 days) with short wash-out periods (30 and 50 days) in between. It is quite possible that the deviations from other studies exist due to some methodological variation. It is also possible that this feeding system led to dysregulation of mineral excretion. In practice, such feeding schedules do occur occasionally and should be discouraged. Regardless of the cause, this finding proves the value of the Lucas test’s data presentation. When examining only the data on apparent digestibility or renal excretion, no clear effect of the feeding regimen was visible in the aforementioned study. The results were not remarkable on their own. However, by plotting the data in the present study, it was demonstrated that the aforementioned feeding systems may indeed have effects on mineral metabolism in ponies.

The well-known effects of acidifying or alkalizing rations on renal excretion and serum Ca concentration could also be shown with these methods. The same was true for the effects of an unbalanced Ca:P ratio on renal excretion. In this context, the absence of effects on faecal excretion of Ca and P from either the acid–base balance or Ca:P ratio is an important finding. In other species, particularly carnivores, imbalanced Ca:P ratios in the diet affect apparent digestibility very much [21,75,76,77,78]. This might be due to the formation of insoluble Ca-P salts. Cehak and Breves [79] suggested that, in horses, Ca is actively absorbed from the small intestine to make P available for the microbiota in the hindgut. This might explain why Ca and P in horses do not negatively affect each other’s digestibility in the same way as in carnivores.

### 4.2. Distinction between Horses and Ponies

In the present study, ponies and horses were differentiated based on body weight (ponies < 300 kg; horses > 300 kg). This is a rather rough method. An individual may fall into different groups when losing or putting on weight. With a limit of 300 kg BW, it is unlikely that a horse would be classified as a pony, but it is quite likely that some ponies might be categorized as horses. In retrospective studies in a meta-analysis, it is not possible to obtain data on the breeds unless they are given in the study. This would reduce the amount of data considerably. It would be even more effective if the classification could also consider breeds known for being easy keepers, such as certain draft horse types or Andalusians (PRE), or be based on wither height.

Despite the aforementioned considerations, ponies had lower endogenous faecal Mg losses compared to horses. A possible explanation might be that most ponies originate from relatively cold regions. Plants grown at lower temperatures usually have lower Mg content than plants grown at higher temperatures [80]. This might force the ponies to accumulate Mg. Other effects such as a higher Ca retention in ponies or a potentially higher digestibility at low intake were not sufficiently clear-cut.

### 4.3. Metabolic Body Weight as Calculation Basis and Differences between Horses and Ponies

In the present meta-analysis, all data are presented per kg of MBW. In species with widely varying adult body weights such as horses or dogs, this has been discussed extensively [24,81,82,83]. To ensure accuracy, all data were also plotted per kg of body weight (BW) where applicable. There was no difference in the results, except for apparent Ca digestibility. For Ca, the difference in apparent Ca digestibility between ponies and horses at low intake was not measurable when intake was expressed as mg/kg BW instead of per kg MBW. The comparison of apparent Ca digestibility was re-evaluated in horses and ponies at an intake level below requirements, in this case, the requirements of NRC 2007 [84], i.e., below 40 mg/kg BW. The data distribution was somewhat different (14 ponies and 10 horses compared to 15 ponies and 8 horses) because requirement limits were either given per kg MBW or kg BW. The apparent digestibility was still numerically different, but the difference was not statistically significant (pony: −9.8%; horse: −26.2%; *p* = 0.095). It remains unclear whether the observed difference between ponies and horses in apparent Ca digestibility at low intake is due to the dimension or is simply a result of data distribution. In contrast, the aforementioned differences between horses and ponies in apparent Mg digestibility remained significant, regardless of the method of plotting or the dimension (Figure 13 and Figure 14).

### 4.4. Effect of Mineral Sources on Mineral Metabolism

When the apparently digested amount of Ca was plotted against intake, it became evident that the regressions for “organic” or “inorganic” sources were different, with a higher slope for the “organic” sources, suggesting a higher true digestibility. Ca in grass is mainly soluble [85]. The solubility of Ca in grass or legumes amounts to 50 to 80% [86]. The “inorganic” sources were mostly CaCO_3_ and CaHPO_4_. These compounds are not very soluble in water. In horses, Ca is mainly absorbed in the small intestine [79]. Therefore, it is not surprising that highly soluble sources have a higher apparent digestibility than Ca compounds with low water solubility.

When the regression between Ca intake and apparently digested Ca was calculated across all data, the resulting equation aligned with the findings of Kienzle and Burger [1]. The slope of the regression equation multiplied by 100 represents the true digestibility. The intercept shows the endogenous losses. The endogenous losses divided by the slope equal the Ca that is required to replace faecal losses (Table 1). The intercept, slope and the required Ca to replace faecal losses diverge between Ca from “organic” and “inorganic” sources (Table 1). This can have consequences in practical feeding. Here, “organic” Ca primarily originates from forages. Thus, horses consuming forage-based rations are more likely to tolerate a marginal Ca intake than horses being fed cereal-based supplemented rations, provided the forage does not derive from exceptionally high oxalate plants. Ca from oxalate is known to be poorly available for horses [87,88], and therefore, Ca requirements increase with high levels of oxalate [89].

Serum Ca levels did not exhibit any effects related to the source or amount of intake, nor did they differ between ponies and horses. There was no difference between postprandial and preprandial sampling time. The only factors that influenced serum Ca levels were aluminium in the diet and the manipulation of acid–base balance in the excluded studies. It is very well known that serum Ca concentration is tightly regulated [90,91,92,93,94] and this has been reconfirmed.

Renal Ca excretion showed no significant effects of being a pony or a horse or of the Ca source that was used. Kienzle and Burger [1] postulated a broken-line model between Ca intake and renal excretion with a breakpoint at about 394.3 mg/kg MBW Ca intake. In the present meta-analysis, no clear-cut breakpoint was identified, and the relationship between intake and renal excretion was not particularly strong. However, in the present study, the limitation for the Ca:P ratio was 1:1 to 5:1. By contrast, Kienzle and Burger [1] limited their data to a Ca:P ratio between 1:1 and 2:1. If the data of the present meta-analysis were similarly limited to a Ca:P ratio between 1:1 and 2:1, they showed a curvy linear relationship between Ca intake and renal excretion rather than a broken-line model (Figure 18). For practical feeding, it is important to note that renal Ca excretion begins to increase substantially at an intake of around 300 mg/kg MBW. This intake is approximately twice the required amount.

Retention of Ca can be caused by reduced bone turnover [95]. If so, it is mandatory that P is also retained. The data showed a weak correlation between Ca and P retention (r = 0.374). It is also obvious that errors in the collection of faeces and urine, as well as in the measurement of food, typically result in an overestimation of retention. Therefore, a correlation between Ca and P retention is to be expected. The retention may as well be caused by Ca and/or P accumulation in the gut or bladder. In that case, Ca retention can be independent of P retention. Therefore, Ca and P retention should not be overinterpreted. Nevertheless, it appears that ponies retain more Ca and P compared to horses. A possible explanation is that ponies, being easy keepers [96], retain more Ca and P to provide for their needs during periods of low food intake. The impact of P solubility on Ca and P retention could not be clearly determined due to the distribution of the data.

Water-soluble P showed a higher apparent digestibility than all other P sources. The same issue applied to true digestibility, particularly at high intake (Figure 8). This corresponds with a higher P serum level, in most cases even above the reference range, as well as with an increased renal P excretion. Similar effects have been reported in other species such as humans, dogs and cats [2,17,18,19,20,97]. Here, the resulting hyperphosphatemia and hyperphosphaturia are considered a potential health risk [5,14,98]. Currently, it is not recommended to use water-soluble “inorganic” P sources for horses and ponies.

Given this recommendation to avoid water-soluble “inorganic” P sources, true digestibility and faecal endogenous losses should only be calculated for other P sources that can potentially be used for equines. The resulting true digestibility of P was lower than the calculations of Kienzle and Burger [1], which included water-soluble “inorganic” P sources (Table 1). The resulting P requirement would be 1.5 times higher than the current recommendations.

The apparent digestibility of Mg was not affected by any one source. This was expected because the “inorganic” Mg was in more than 50% of the studies MgO, which is known to be highly available [99]. Mg from leaves is highly available as well, even in humans [100], whose digestive capacity for plant material is much lower than that of horses. When plotting faecal Mg excretion against intake in a modified Lucas test, the true digestibility for Mg matched the results of Kienzle and Burger [1] and so did the endogenous losses across all data from ponies and horses. As mentioned above, the true Mg digestibility did not differ between ponies and horses, but the endogenous Mg losses were lower in ponies. Therefore, a lower Mg requirement for ponies could be postulated (Table 1).

The distribution of serum Mg levels did not permit an evaluation of differences between ponies and horses. In the overlap area, where data from both ponies and horses were available, no difference could be observed. The slight increase in serum Mg content with increasing Mg intake supports the well-known fact that Mg blood levels can be used to estimate Mg supply [94]. An exception was the data by Pferdekamp [50], which showed comparatively low serum Mg levels despite high Mg intake. One possible explanation for this finding is the extremely high intake of Ca and P in this study. It is known that high levels of Ca and P reduce the availability of Mg [61,101,102] and lead to lower serum Mg levels [103].

Renal Mg excretion increased with higher Mg intake (y = 0.3308x − 0.2206; r^2^ = 0.773) as well as with higher apparently digested Mg (Figure 16). This concurs with previous research [104]. There was very little Mg retention, and it was independent of intake. However, there was a weak correlation between Ca retention and Mg retention, as well as between P retention and Mg retention, particularly when both values were greater than zero (Ca/Mg retention: r = 0.498; P/Mg retention: r = 0.606). This finding suggests that the aforementioned errors during collection may indeed play a role. Retention data should be interpreted with great caution.

## 5. Conclusions

Water-soluble “inorganic” P sources such as Na_2_HPO_4_, NaH_2_PO_4_ or even Ca(H_2_PO_4_)_2_ are more readily available than other “inorganic” and “organic” P sources. Water-soluble “inorganic” P leads to an increase in both serum P levels and renal excretion. In other species, this has been shown to pose a health risk. Including data from experiments with water-soluble P sources into the regression calculations for true digestibility may subsequently lead to an underestimation of factorially calculated P requirements in horses being fed water-insoluble P.

In the Lucas test, the true digestibility of Ca from “organic” sources was higher than from “inorganic” sources. Thus, horses consuming forage-based rations are more likely to tolerate a marginal Ca intake than horses deriving Ca from mineral salts.

True digestibility of Mg was the same in horses and ponies. But endogenous Mg losses were significantly higher in horses than in ponies. This finding suggests a lower requirement for ponies compared to horses.

## Figures and Tables

**Figure 1 animals-14-02765-f001:**
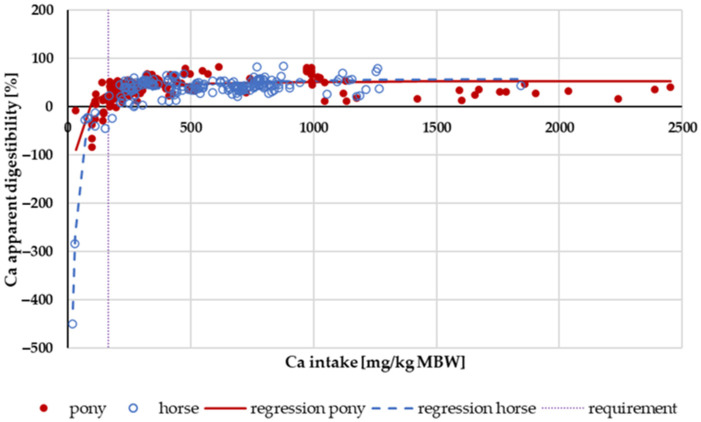
Relationship between Ca intake (in mg/kg MBW) and apparent Ca digestibility (in %). Regression lines mark data of horses compared to ponies.

**Figure 2 animals-14-02765-f002:**
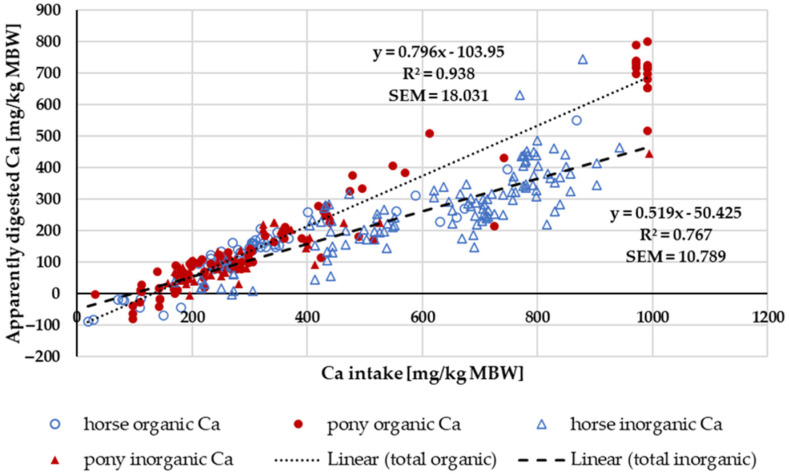
Relationship between Ca intake and apparently digested Ca (both in mg/kg MBW). Trendlines mark “organic” and “inorganic” Ca sources. Endogenous losses are represented by the intercept and true digestibility by the regression coefficient. Only data up to an intake of 1000 mg/kg MBW and with information on Ca source were taken into account.

**Figure 3 animals-14-02765-f003:**
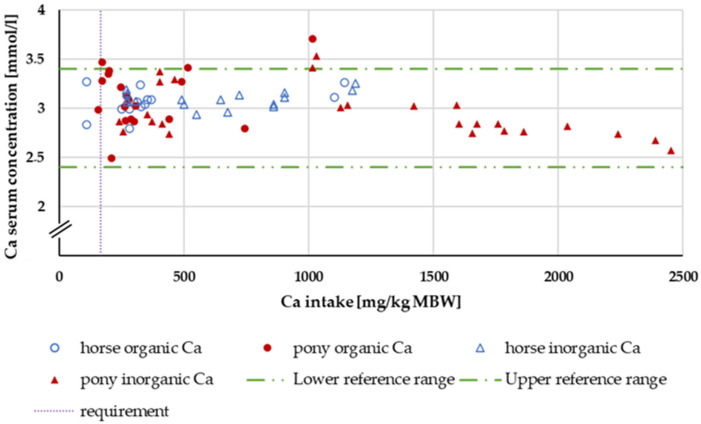
Serum Ca concentration (in mg/kg MBW) plotted against Ca intake (in mmol/L). Data without information on Ca source were not used. Experiments with shifts in acid–base balance and with addition of aluminium were excluded.

**Figure 4 animals-14-02765-f004:**
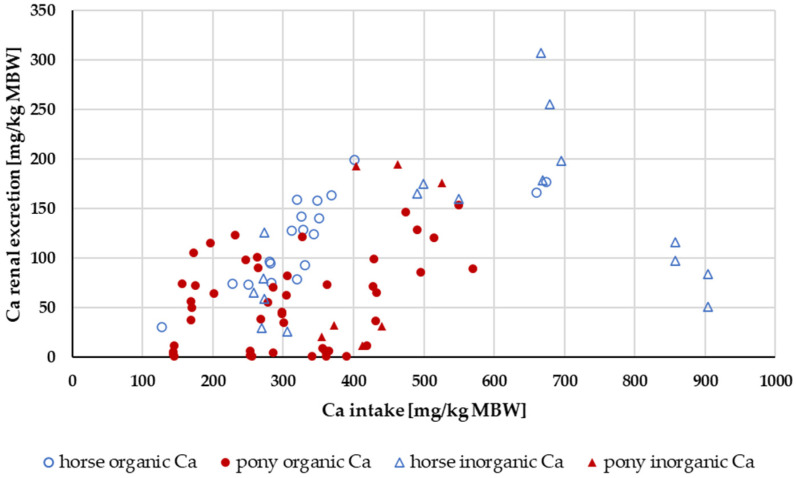
Relationship between Ca intake and renal Ca excretion (both in mg/kg MBW). Only data up to an intake of 1000 mg/kg MBW and with information on Ca source were taken into account. The Ca:P ratio was limited to 1:1–5:1. Experiments with shifts in acid–base balance were excluded.

**Figure 5 animals-14-02765-f005:**
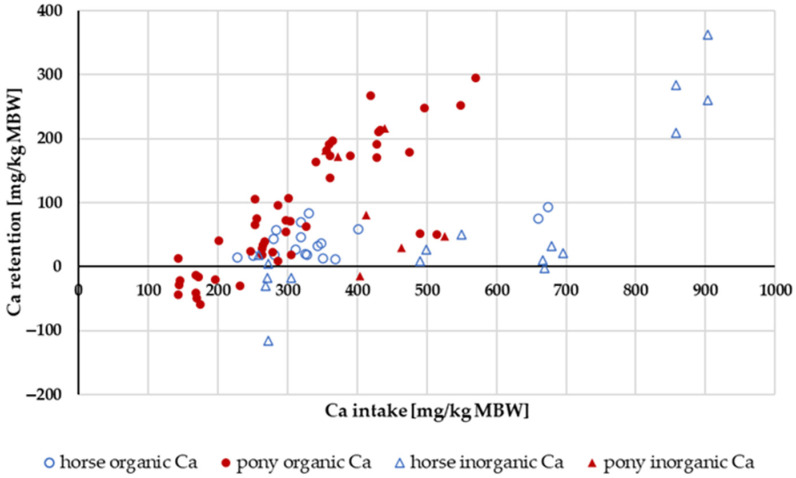
Ca retention plotted against Ca intake (both in mg/kg MBW). Only data up to an intake of 1000 mg/kg MBW and with information on Ca source were taken into account. The Ca:P ratio was limited to 1:1–5:1. Experiments with shifts in acid–base balance were excluded.

**Figure 6 animals-14-02765-f006:**
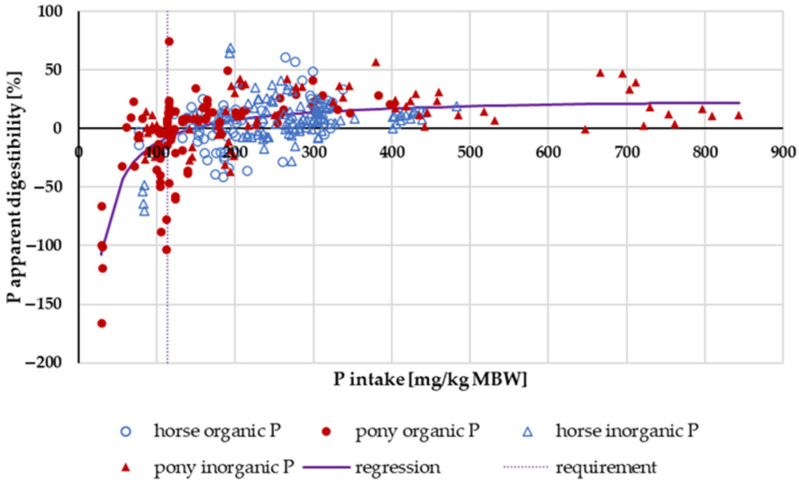
Relationship between P intake (in mg/kg MBW) and apparent P digestibility (in %). Data without information on P source were not used.

**Figure 7 animals-14-02765-f007:**
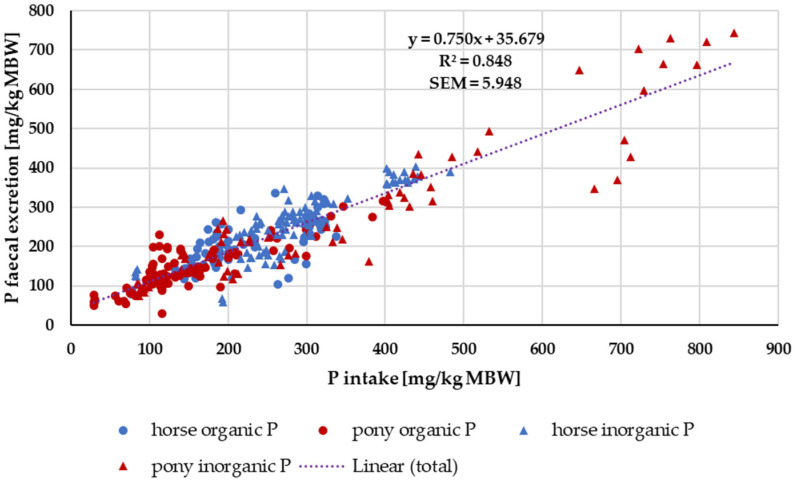
Faecal P excretion in relation to P intake (both in mg/kg MBW). Data without information on P source were not used.

**Figure 8 animals-14-02765-f008:**
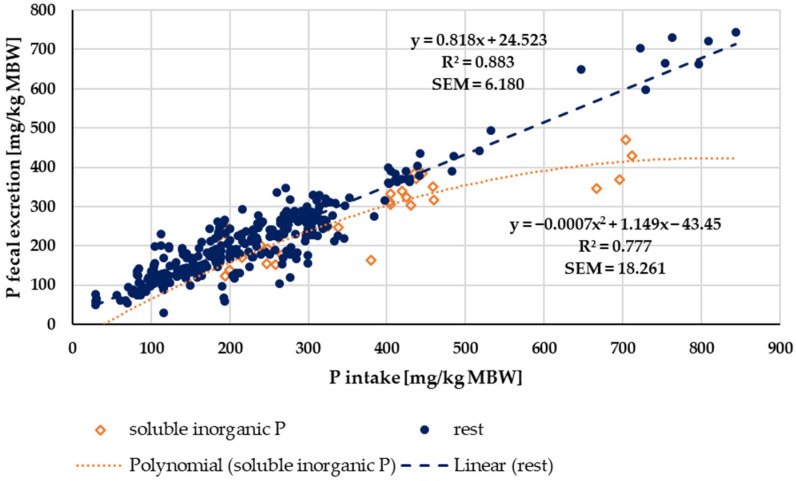
Faecal P excretion in relation to P intake (both in mg/kg MBW). Data without information on P source were not used. Trendlines mark water-soluble “inorganic” P sources and all other sources.

**Figure 9 animals-14-02765-f009:**
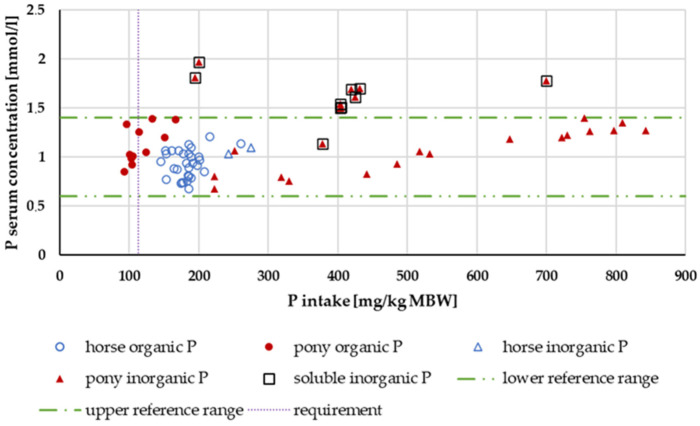
Serum P level (in mg/kg MBW) plotted against P intake (in mmol/L). Data without information on P source and with shifts in acid–base balance were not used.

**Figure 10 animals-14-02765-f010:**
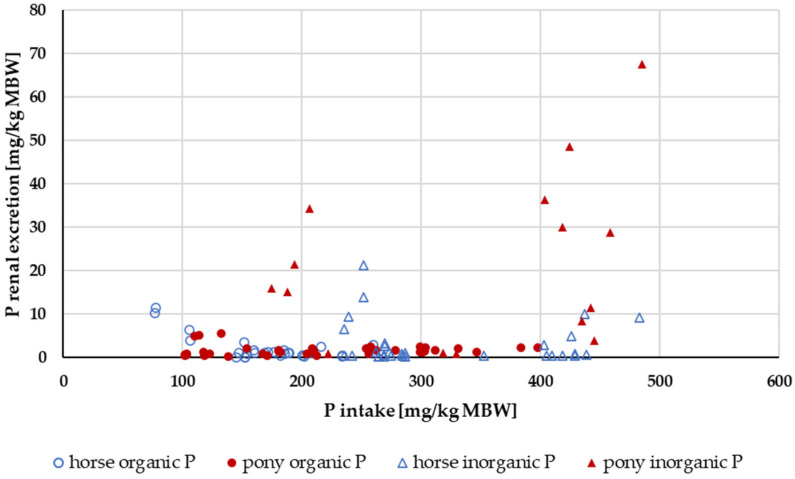
Relationship between P intake and renal P excretion (both in mg/kg MBW). Only data up to intake of 500 mg/kg MBW and with information on P source were taken into account. Ca:P ratio was limited to 1:1–5:1. Experiments with shifts in acid–base balance were excluded.

**Figure 11 animals-14-02765-f011:**
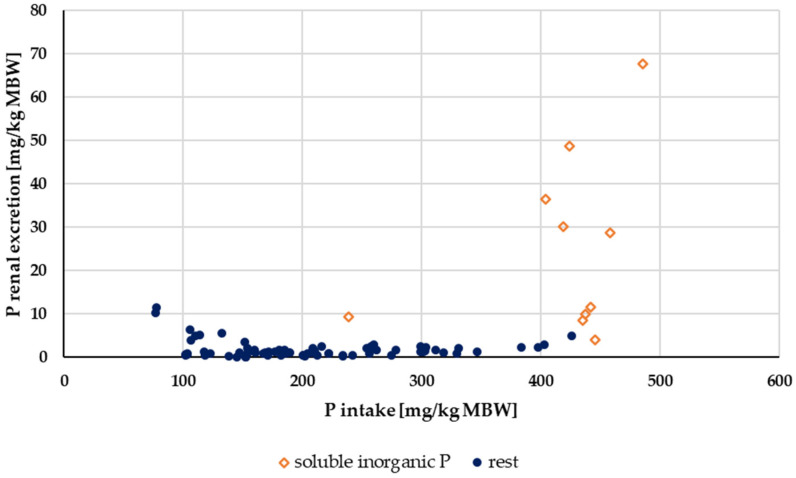
Relationship between P intake and renal P excretion (both in mg/kg MBW) from Figure 10 with differentiation of P sources into soluble “inorganic” sources and other sources. Data on unknown “inorganic” P sources were not used.

**Figure 12 animals-14-02765-f012:**
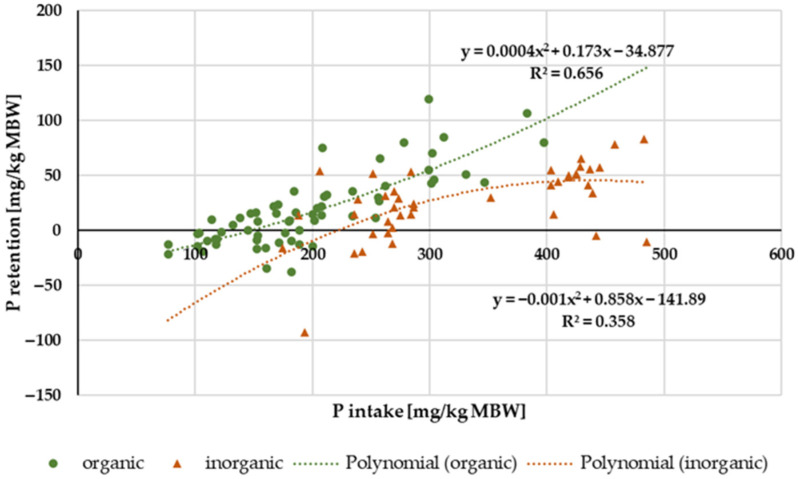
Relationship between P intake and P retention (both in mg/kg MBW). Only data up to intake of 500 mg/kg MBW and with information on P source were taken into account. Ca:P ratio was limited to 1:1–5:1. Experiments with shifts in acid–base balance were excluded. P sources were differentiated into “organic” and “inorganic” sources.

**Figure 13 animals-14-02765-f013:**
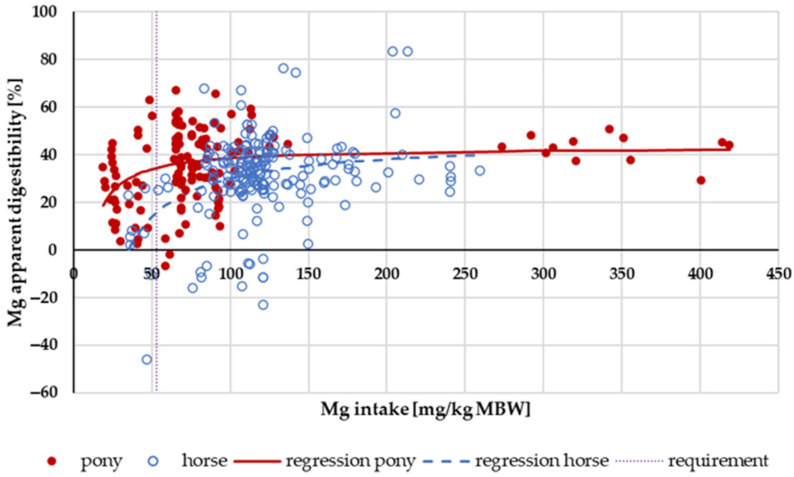
Relationship between Mg intake (in mg/kg MBW) and apparent Mg digestibility (in %). Regression lines mark data of horses compared to ponies.

**Figure 14 animals-14-02765-f014:**
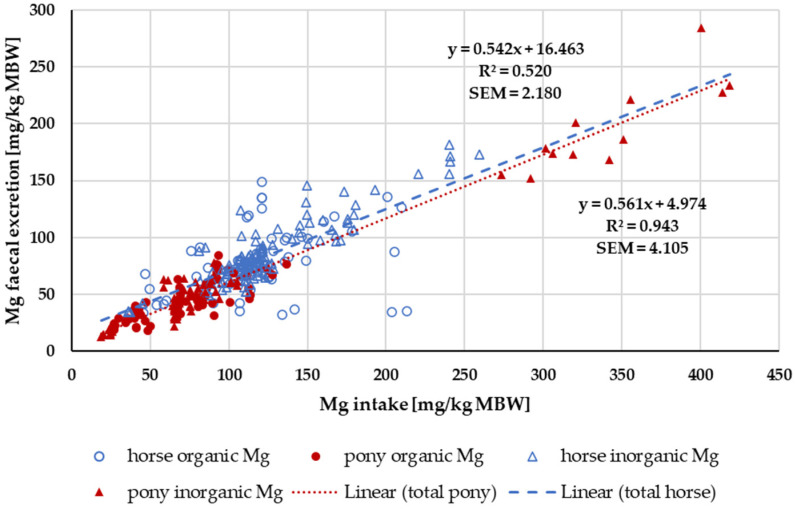
Relationship between Mg intake and faecal Mg excretion (both in mg/kg MBW). Data without information on Mg source were not used. Trendlines mark data of horses compared to ponies.

**Figure 15 animals-14-02765-f015:**
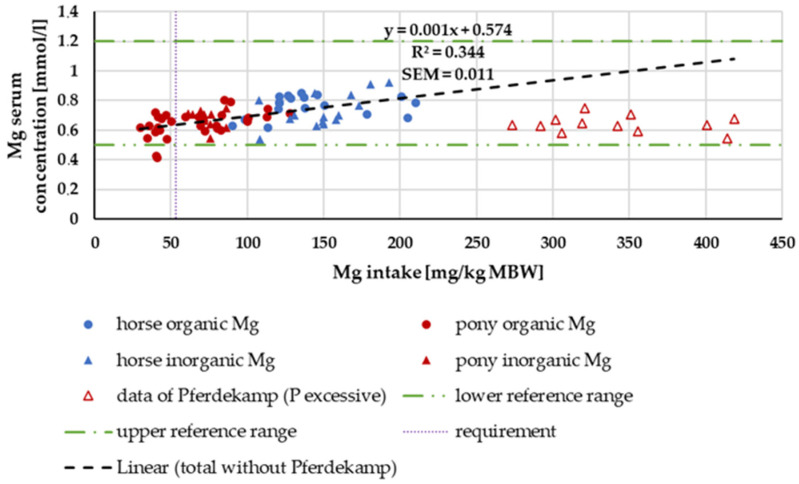
Serum Mg (in mg/kg MBW) plotted against Mg intake (in mmol/L). Data without information on Mg source were not used. Trendline marks serum Mg content without data from Pferdekamp.

**Figure 16 animals-14-02765-f016:**
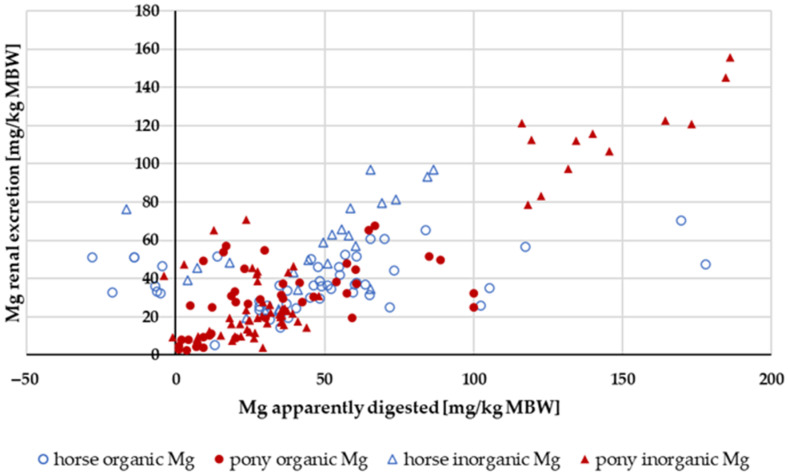
Renal Mg excretion in relation to apparently digested Mg (both in mg/kg MBW). Data without information on Mg source were not used.

**Figure 17 animals-14-02765-f017:**
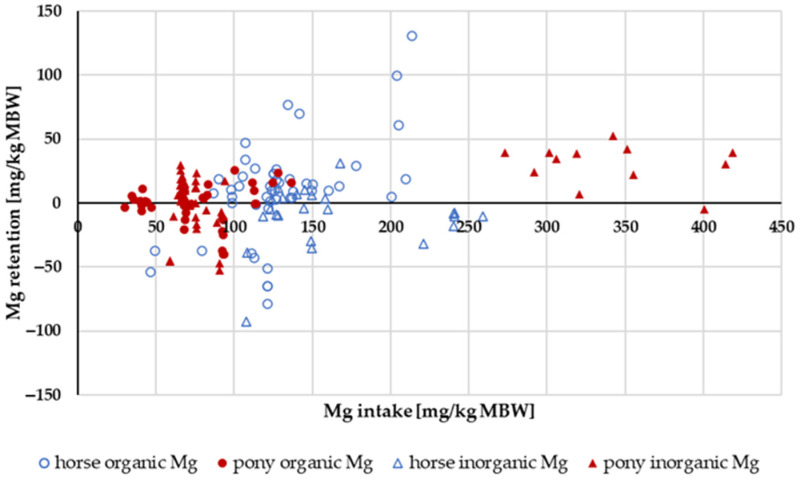
Relationship between Mg intake and Mg retention (both in mg/kg MBW). Data without information on Mg source were not used.

**Figure 18 animals-14-02765-f018:**
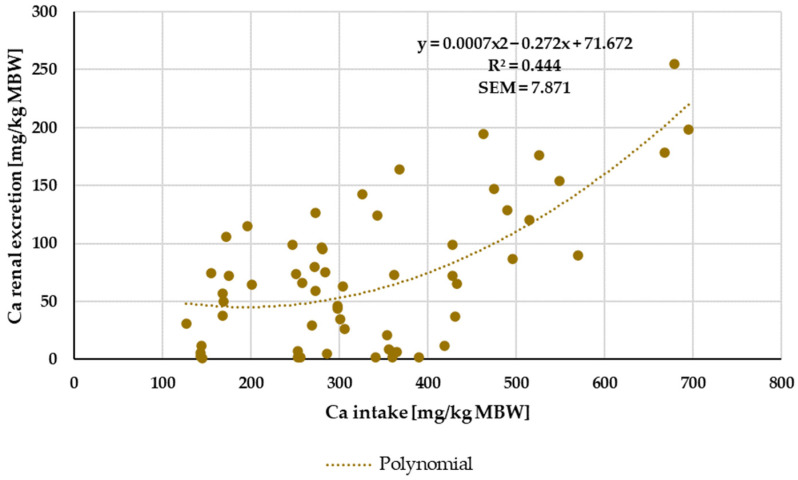
Relationship between Ca intake and renal Ca excretion (both in mg/kg MBW). Only data up to intake of 1000 mg/kg MBW were taken into account. Ca:P ratio was limited to 1:1–2:1. Experiments with shifts in acid–base balance were excluded.

**Table 1 animals-14-02765-t001:** Comparison of the results of the present study with those of Kienzle and Burger.

Study and Topic	Ca	P	Mg
True digestibility [%] following Kienzle and Burger	59	28	46
True digestibility [%] in the present study	All data: 65	All sources except water-soluble “inorganic”: 18	All data: 45
	“Organic”: 80		Pony: 44
	“Inorganic”: 50		Horse: 46
Endogenous losses [mg/kg MBW] following Kienzle and Burger	39	26	9
Endogenous losses [mg/kg MBW] in the present study	All data: 77	All sources except water-soluble “inorganic”: 25	All data: 11
	“Organic”: 50		Pony: 5
	“Inorganic”: 103		Horse: 17
Mineral requirements [mg/kg MBW] to replace faecal losses following Kienzle and Burger	≙39/59 × 100 = 66	≙26/28 × 100 = 93	≙9/46 × 100 = 20
Mineral requirements [mg/kg MBW] to replace faecal losses in the present study	All data ≙ 77/65 × 100 = 118	≙25/18 × 100 = 139	All data ≙ 11/45 × 100 = 24
	“Organic” ≙ 50/80 × 100 = 63		Pony ≙ 5/44 × 100 = 11
	“Inorganic” ≙ 103/50 × 100 = 206		Horse ≙ 17/46 = 37

## Data Availability

Data presented in this study are available upon request from the corresponding author.

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
