# Peer review of "A Meta-Analysis on Quantitative Calcium, Phosphorus and Magnesium Metabolism in Horses and Ponies"

_animals, 2024, doi:10.3390/ani14192765_

Round 1

Reviewer 1 Report

Comments and Suggestions for Authors

Thank you for a nice manuscript and a well-performed and described meta-analysis. I have some small comments and suggestions.

Material and methods

Did you consider age? Training? Pregnancy and lactation? I think you also have to clarify the inclusion/exclusion criteria for those parameters.

Row 119 to 122. Avoid using personal pronouns as she, her.

Discussion

Line 319 Should it be considered? 

Reviewer 2 Report

Comments and Suggestions for Authors

This study conducts a meta-analysis to update and expand on previous research, focusing on the calcium, phosphorus, and magnesium metabolism in horses and ponies. The article effectively addresses the three questions they raised. The data was carefully screened and the results that are clinically relevant. However, there are a few minor issues. There are too many figures, making the content somewhat cumbersome. It is recommended to present the data more intuitively and include confidence intervals for the data. Revisions are necessary to improve the clarity and rigor of the findings before the manuscript can be considered for publication.

Abstract:

1.       The terminology should maintain consistency like “serum P” and “serum P levels” in the article. Please also review the other terminology; it is best to maintain consistency throughout.

Introduction:

2.       Suggest changing the spelling of data base to database. (Line 51.)

2. Materials and Methods

3.       In the original text “google scholar” should be changed to “Google scholar”. (Line 65.)

4.       It is best to provide a brief introduction to the Lucas-test and the modified Lucas-test. (Line 103-104)

5.       The “0,05” should be “0.05”.  (Line 154)

6.       Although the literature search section is quite detailed, it is recommended to include the number of articles reviewed and to provide a brief description of the animals used in the studies.

3. Result:

7.       Figure 4 and Figure 5 are different in size from the other figures and are not clear enough. (Line 201, Line 205)

8.       3Is this "higher" significant? It is recommended to provide a p value and 95% confidence interval. (Line 279-280)

9.       Can it be mentioned whether there are any differences between horses and ponies, as well as between organic and inorganic sources? (Line 288-293)

10.    The style of the line showing the Mg requirement in Figure 15 is different from that in the figure note. (Line 294)

11.    The vertical axis of the graph is misaligned. (Line 309)

4. Discussion

12.    Limiting the study to adult animals in maintenance or work conditions without discussing the applicability of the findings to other life stages (e.g., growth, pregnancy, lactation) is a limitation.

13.    The “2” should be a superscript, and there are several others in the text; please check carefully. (Line 432)

Reviewer 3 Report

Comments and Suggestions for Authors

Dear authors,

Thank you for submitting an interesting and thorough review on equine calcium, phosphorus and magnesium metabolism. The article is of interest to the circles of equine nutritionists, physiologists and medicine specialists, and methodologically convincing. Despite providing valuable insights on equine electrolyte metabolism, it requires extensive language corrections and stylistic improvements. The remaining comments will be structured by section.

Simple summary and Abstract

- l. 10 the definition of organic and inorganic may not be this relevant to the simple summary
- l. 20 aim II.: regarding requirements as well
- l. 25 the sentence is unclear, what which means were compared?
- l. 28 "current requirements" do the authors mean current recommandations? By the way, the recommandations the authors refer to is unclear at this stage. Maybe say "the GfE's recommandations"
- l. 37 "The intake of these P sources led to hyperphosphatemia and hyperphosphaturia." Is this effect dose-dependent? The current formulation suggests that water soluble P cannot be retained in the organism and is necessarily excessive.

Introduction

- The relevance of Ca, P and Mg metabolism to equine health may explained briefly

Methods

- As stated in the instructions for authors, Animals requires meta-analyses and systematic reviews to be conducted in accordance to the PRISMA statement (https://systematicreviewsjournal.biomedcentral.com/articles/10.1186/s13643-021-01626-4). The authors describe the search methodology in detail and should have no issues in implementing the recommended methodology in the manuscript. Should the authors have used the PRISMA guidelines, they should state so explicitly.
- l. 72 "Only experiments with adult animals in maintenance or during 72 work were used". Since animals at work were also considered, could the authors comment on the losses or expected losses by sweat? (possibly in the discussion)
- l. 82 where the breeds always unavailable or what justifies this classification based on body weight?
- l. 98 Handbuch Pferdepraxis, is there a primary source available?
- l. 116-122 please define the methods and criteria for outlier detection here and the results of outlier analysis in the appropriate results section
- l. 128-130, l. 136-143 please provide these details in the results and not in the methods section
- l. 146 normal distribution of outliers is expected for all mentioned statistical test but the Mann-Whithney U. Were the residuals checked for normality and homoskedasticity/sphericity? Some plots look like robust regression may be a better choice.

Results

- The data sources should be named in the respective sections and include information on excluded data (as commented above). This may be done as text or with tables (providing sample sizes (separately for horses and ponies) and details deemed relevant for interpretation by the authors).
- l. 161 "The Ca requirement for maintenance is 164 mg/kg MBW" is this the recommendation of the GfE or the calculated requirement from this meta-analysis?
- l. 166: "The regression coefficient was higher for “inorganic“ sources." The opposite seems true (see Figure 2 or compare with l. 169).
- l. 167 did the two way ANOVA include an interaction between factors? If yes, please mention the coefficient and p-value.
- Figure 2: please develop the description of the results from figure 2 (true digestibility, endogenous losses)
- l. 179: the authors probably mean "Serum Ca concentration"? (same is true for later subsections)
- Figure 3: The horizontal lines (reference range etc) are difficult to distinguish from the grid lines. Moreover, the range of the y-axis appears unnecessarily large.
- l. 187: Ca/P -> Ca:P (to match the ratios given afterwards)
- Figure 4 and 5 are too small
- Figure 4: the data on horse inorganic Ca describes a parabolic shape which is also partly seen in the horse organic Ca data. This indicates that ANOVA may not be adequate to analyse the data. Can the authors provide a tentative explanation for this phenomenon? Could differences in the Ca:P ratio explain the lower calcium excretion under very high Ca intake?
- l. 218 please provide units
- Figure 8: please spell out polynomial
- Figure 12: could the authors provide regression lines and further statistical analysis or explain why they considered those unnecessary?

Discussion

- l. 337 "It is quite possible that the deviations to other studies are due to some systematic experimental fault." The authors have just provided a possible explanation for these deviations (shift from excess to deficiency during the experiment), as a result it appears harsh to qualify the experiments a systematically faulty. The qualification of "methodological variation" appears more prudent.
- l. 382 I suggest providing the data supporting this claim as an additional file.
- l. 416-425 see comments on Figure 4
- l. 432-433 please provide a reference for this statement
- l. 473 please round the values to an experimentally justified number of decimals (i.e., corresponding to the experimental precision).
- l. 476-478: correlation is usually given as r or rho (Pearson or Spearman's correlation coefficient), not as r²
- Additional limitations may be discussed. E.g., several unaccounted determinants of Ca and P digestigility (ratio not included in the calculation, hormonal effects, possibly seasonal effects, age, sex, ...). I assume lactating mares were not included?

Conclusion

- The conclusions could be expanded to include more of the main results of the manuscript.

Comments on the Quality of English Language

Dear authors,

The following comments are by no means exhaustive, as extensive language corrections are needed. In general, please try to keep the sentences straightforward, spell out abbreviations on first use, and check the punctiation marks.

Simple summary & Abstract

- l. 9 looking at -> investigating
- l. 10 ponies and "organic" -> ... ponies regarding "organic" ...
- P, Ca and Mg must should be defined (i.e., spelled out at least once) before using the abbreviation. Even if these abbreviations are common in nutritionist circles, they may not be known by all Animals readers.
- l. 12-13 please try to keep the sentences straightforward. "Horses appear to have substantially (alternative: four times) higher Mg requirements than ponies."
- l. 14 "For Ca, there were differences between “inorganic” and “organic” sources
with a higher availability of Ca from plant origin." redundant sentences, shorten to "Organic Ca was shown to have a higher bioavailability than inorganic Ca." Similar for P
- l. 15 "For the P sources" missing a ,
- l. 25 lack of punction (,) makes some sentences difficult to read
- l. 35 avoid double negation "no unequivocal"

Introduction

- l. 47 mean-time -> meantime
- l. 49 bigger -> larger
- l. 49 "data base"?

Methods

- l. 97 GfE please spell out abbreviations on first use
- l. 104 in *the* Lucas test
- l. 105 provided *that the* data distribution, drop calculation
- l. 114 concentration and not content was plotted
- l. 125 "did not stick out" please avoid familiar language
- l. 126 *which* represented outliers?
- l. 128 O'Connor not O`Connor
- l. 148: please replace "with the posthoc test All Pairwise Multiple Comparison
Procedures (Holm-Sidak method)" with "followed by the Holm-Sidak posthoc test for all pairwise comparisons"
- l. 153 please provide all numerical values with . and not , as decimal separator

Discussion

- l. 343 "The plots, however, showed that something was different in her study." Unspecific and poorly formulated.
- l. 353: each-others -> eachother's
- l. 355 distinction may be a better choice than "Discrimination"
- l. 374 "extensively" rather than "intensively"
- l. 374 "to make sure" of what?
- Table 1: following Kienzle and B, not "after"

Round 2

Reviewer 2 Report

Comments and Suggestions for Authors

The methodology has been clarified, and the statistical models have been properly explained. The use of older references has been sufficiently justified, especially given the stability of muscle physiology over time and the limited recent research in this specific field. Additionally, while the absence of more recent data could be considered a limitation, it does not detract from the robustness of the findings, and the thorough analysis provided makes this manuscript a valuable contribution to the literature.

I have no further suggestions for revision, and I recommend that the manuscript be accepted for publication.

Author Response

Dear reviewer,

thank you for your response. We are pleased that we were able to implement your suggestions to your contentment.

Kind regards

Reviewer 3 Report

Comments and Suggestions for Authors

Dear Authors,

Thank you for submitting a much improved manuscript and for providing additional information to illustrate your response. The vast majority of my comments have been addressed. The only remaining issue is compliance with the PRISMA statement. As this is a requirement of the journal and goes beyond my personal opinion, I leave it to the editor to decide whether the manuscript is acceptable in its present form. However, please note that this statement is enforced by all major publishers and can be considered a consensus in the scientific community. The authors have already been kind enough to provide the PRISMA flowchart for the literature sources under consideration. A compromise might be to mention the PRISMA statement in the Methods section and provide the flowchart and any other information required by the PRISMA checklist that is not included in the manuscript in an additional file. The reason for this is that searches may include non-peer-reviewed material (e.g. online) that is subject to change, or errata may be published to correct previous publications. The aim of the checklist is to ensure that the systematic review (like any other scientific study) is reproducible; if the date is missing and a publication is changed (e.g. via an erratum), this is not transparent to readers or subsequent researchers. A simple solution to this problem would be to state that all resources were last accessed in June 2024 (or whatever month the manuscript was written).

As a final and very minor comment, "Student" in Student's t-test is the pseudonym of William Sealy Gosset and should therefore be capitalised.

Kind regards.

Author Response

Dear Reviewer,

thank you for suggesting a simple solution to the PRISMA problem. We did not quite understand the problem in round 1. We used relatively old literature. Most authors are no longer in science (if alive). Therefore changes of errata are not a likely event. Writing down the time of last assessment is an elegant solution, thank you. We are very grateful that you realised that due to a misunderstanding we did not list several of our sources in the references. We have changed that. We also have made the flow diagram and a table available as supplementary material.

Kind regards.